# Linking ROS Levels to Autophagy: The Key Role of AMPK

**DOI:** 10.3390/antiox12071406

**Published:** 2023-07-10

**Authors:** Francesco Agostini, Marco Bisaglia, Nicoletta Plotegher

**Affiliations:** 1Department of Biology, University of Padova, Via Ugo Bassi 58/B, 35131 Padova, Italy; 2Study Center for Neurodegeneration (CESNE), 35121 Padova, Italy

**Keywords:** AMPK, autophagy, lysosomes, mitochondria, ROS

## Abstract

Oxygen reactive species (ROS) are a group of molecules generated from the incomplete reduction of oxygen. Due to their high reactivity, ROS can interact with and influence the function of multiple targets, which include DNA, lipids, and proteins. Among the proteins affected by ROS, AMP-activated protein kinase (AMPK) is considered a major sensor of the intracellular energetic status and a crucial hub involved in the regulation of key cellular processes, like autophagy and lysosomal function. Thanks to these features, AMPK has been recently demonstrated to be able to perceive signals related to the variation of mitochondrial dynamics and to transduce them to the lysosomes, influencing the autophagic flux. Since ROS production is largely dependent on mitochondrial activity, through the modulation of AMPK these molecules may represent important signaling agents which participate in the crosstalk between mitochondria and lysosomes, allowing the coordination of these organelles’ functions. In this review, we will describe the mechanisms through which ROS activate AMPK and the signaling pathways that allow this protein to affect the autophagic process. The picture that emerges from the literature is that AMPK regulation is highly tissue-specific and that different pools of AMPK can be localized at specific intracellular compartments, thus differentially responding to altered ROS levels. For this reason, future studies will be highly advisable to discriminate the specific contribution of the activation of different AMPK subpopulations to the autophagic pathway.

## 1. Introduction

Reactive oxygen species (ROS) comprise a set of molecules that derives from the incomplete reduction of oxygen [1]. Although ROS are frequently considered detrimental factors that mediate toxic effects and are often associated with the onset and progression of several human diseases and with aging, they also represent physiological bioproducts of cellular respiration. Therefore, they participate, as important signaling entities, in the regulation of several cellular processes, including cell differentiation, cell growth, and apoptosis [2]. The concentration of ROS is tightly controlled and highly depends on the rate of their production and degradation. When the intracellular amount of these molecules exceeds the physiological range, they can perturb cell homeostasis, leading to so-called oxidative stress and oxidative damage [3]. Indeed, ROS function is based on their high reactivity, which makes them able to interact with all kinds of biomolecules, including DNA, lipids, and proteins. Through the oxidation of cell components, ROS can promote the modification of their targets, affecting their functions [4]. For this reason, analyzing in detail the influence that ROS may have on other molecules is of great relevance, not only for the understanding of cell physiology but also for the investigation of their involvement in pathological conditions.

Here, we will focus in particular on the effects of ROS on proteins, which comprise posttranslational modifications, variation in the aminoacidic composition, protein structure, subcellular localization, and alterations to protein turnover [5]. The extent and impact of protein modification may depend on the concentration and nature of ROS. The superoxide anions, hydrogen peroxides (H_2_O_2_), and hydroxyl radicals are the most prevalent and the most studied oxidant species. At high intracellular concentrations, ROS can react with all amino acids, but the thiol groups of cysteine residues, methionines, and metal centers appear to be the preferential targets for oxidation [6]. Protein modifications may result in major alterations not only to their function but also to the signaling pathways they participate in. For example, the perturbation of phosphatase and kinase activity may have a high impact on the regulation of their downstream targets, significantly affecting crucial cellular processes [7].

For their ability to react with other intracellular molecules, influencing their concentration and activity, ROS represent key signaling factors, having a crucial role in communications among organelles. In this frame, it is worth noting that the major intracellular source of ROS is represented by mitochondria, which are the most important organelles devoted to aerobic respiration. The production of mitochondrial ROS (mROS) is mainly determined by the leakage of electrons at the level of the mitochondrial electron transport chain (ETC). Indeed, 11 sites involved in substrate catabolism, electron transport, and oxidative phosphorylation have been discovered to be responsible for ROS generation in mammalian mitochondria. Electron leakage from the ETC and their interaction with oxygen to form superoxide radicals or hydrogen peroxide is an event that occurs at low levels in physiological conditions but can be exacerbated under mitochondrial stress and/or mitochondrial malfunction [6].

In light of the aforementioned considerations, it is not surprising that mROS may play a role in the transduction of signals from mitochondria to other cell compartments. Among the different organelles, in the present review, we will focus on the molecular mechanisms involved in the communication between mitochondria and lysosomes, which eventually results in the modulation of the autophagic flux. Special emphasis will be given to the interconnecting role exerted by the AMP-activated protein kinase (AMPK) and to the modulatory activity mediated by ROS, and, in particular by mROS.

## 2. AMPK at the Crosstalk between Mitochondria and Lysosomes

In the last few years, the interaction between mitochondria and lysosomes has gained growing attention, due to the great impact of these organelles in different cellular processes and their crucial involvement in several human pathologies. In fact, it is increasingly clear that mitochondria and lysosomes are important signaling hubs. This notion has led to the development of a novel research field that aims at understanding the mechanisms of long-distance communication between these organelles [8,9].

In this frame, it has been demonstrated that mitochondria can inform lysosomes of their activity, impacting the entire autophagic machinery. More specifically, Raimundo and co-workers demonstrated that mitochondrial impairments result in autophagic alterations, with the accumulation of non-functional lysosomes, the decrease in lysosomal activity, and the reduction in lysosomal pH [10]. Importantly, mitochondrial defects were associated with a decrease in the phosphorylated active form of AMPK. This protein is frequently considered a sensor of intracellular energetic status and participates in the regulation of important cellular processes, such as cell differentiation, cell growth, apoptosis, and autophagy [11]. Upon mitochondrial defects, the chemical activation of the protein was able to reverse the autophagic and lysosomal impairment, confirming the involvement of AMPK in the signal transduction from mitochondria to the autophagic pathway. In addition, AMPK activity was shown to differentially respond to specific mitochondrial insults, since a reduction in protein activation was observed upon chronic mitochondrial defects, while increased activity was detected in the presence of acute mitochondrial impairments [10]. The differential response of AMPK to the alteration in mitochondrial activity likely represents a feedback mechanism to ensure the maintenance of a stable mitochondrial network. Indeed, upon acute mitochondrial insult, the hyperactivation of AMPK and the induction of autophagy could improve the removal of dysfunctional mitochondria through mitophagy. Conversely, in the case of a prolonged mitochondrial impairment, the decrease in autophagic rate could avoid the degradation of the whole mitochondrial network. In light of these data, AMPK can be seen as a crucial mediator of mitochondrial and lysosomal functions, able to influence the dynamics of both organelles and allowing the coordination of their activities. It is worth mentioning that ROS are also produced in other intracellular compartments and via mitochondrial-unrelated intracellular processes, and thus the alterations in their physiological levels are not always associated with mitochondrial defects. Moreover, the source of ROS is not always clearly identified when studying AMPK activation, thus making it more difficult to define this matter in all the examined models.

AMPK is a trimeric kinase complex composed of the catalytic subunit α and the regulatory subunits β and γ [12]. The activation of the protein mainly depends on the phosphorylation of Thr172 in the α subunit, which is frequently used as an indication of protein activity [13]. In mammals, the calcium/calmodulin-dependent protein kinase kinase β (CaMKKβ) and the liver kinase B1 (LKB1) are the most relevant kinases that target AMPK [14]. The ability of AMPK to respond to stimuli deriving from mitochondria is largely based on the fact that the phosphorylation and activation of the protein are highly regulated by nucleotides. Binding with AMP or ADP to the γ subunit leads to the allosteric activation of AMPK, through a mechanism that has not been fully characterized yet. What is known is that AMP and ADP promote the phosphorylation of Thr172 by LKB1, while ATP inhibits AMPK activity, making the protein less prone to phosphorylation [11]. Through this mechanism, AMPK can sense variations in the rate of mitochondrial ATP production with precision and can be tuned in relation to mitochondrial activity.

It is not only nucleotides that can regulate AMPK activity, since the kinase complex is also susceptible to changes in the intracellular concentration of calcium (Ca^2+^). Increased intracellular Ca^2+^ levels stimulate the kinase CaMKKβ, which phosphorylates AMPK. In this way, the sensing of Ca^2+^ may represent an alternative mechanism by which the protein complex allows the transduction of mitochondrial stimuli to other sites of the cells [14].

Another factor that has been reported to modulate the AMPK function is the alteration of ROS concentration. Accordingly, in human skin fibroblasts treated with sub-lethal concentrations of hydrogen peroxide, AMPK activation was observed, associated with an AMPK-dependent increase in the glycolytic flux accompanied by the elevation of intracellular NADPH and GSH [15]. Interestingly, similar results were also observed in fibroblast from Myoclonic Epilepsy and Ragged Red Fibers (MERRF) patients, suggesting that AMPK-mediated metabolic switch and antioxidant response are essential for cell survival in affected tissues harboring a pathogenic mtDNA mutation [15]. However, how variations of ROS levels participate in AMPK activation is still largely unknown. Nevertheless, some pieces of evidence indicate that ROS can modulate AMPK function indirectly, by impacting the relative concentration of nucleotides and calcium levels. Moreover, the direct oxidation of specific amino acids of the α-subunit has also been described [16]. The research papers regarding the role of ROS in the regulation of AMPK are not always concordant, and this may be explained by the fact that the effects of ROS are cell- and tissue-dependent or may rely on the different impact of specific ROS molecules. In the following sections, we will review the proposed mechanism of ROS-mediated modulation of AMPK, discussing the possible reasons for the contrasting results and focusing on the effects of AMPK alteration on important downstream processes, such as autophagy.

## 3. ROS and AMPK: Indirect Regulation

As already mentioned, AMPK activity is influenced by different factors; among them, the variation in ROS levels has been clearly associated with the modulation of the protein function. However, despite more than twenty years of research on this topic the precise mechanisms by which ROS participate in AMPK regulation have not been fully elucidated yet [14].

In one of the first papers showing a positive correlation between ROS levels and AMPK activity, published in 2001, it was observed that in NIH-3T3 cells the exposure to H_2_O_2_ causes a dose-dependent activation of AMPK, as revealed by the increase in phosphorylation at the level of Thr172 in the α subunit [17]. Noteworthily, this effect was not associated with a direct interaction between oxygen species and the protein, but was rather linked to the ROS-mediated increase in AMP:ATP ratio, as oxidative stress was previously shown to reduce the concentration of ATP [18]. Overall, this work demonstrated that ROS can modulate AMPK activity through the AMP-linked allosteric activation of the protein, which induces its phosphorylation by the upstream kinase LKB1. Similar results were obtained also in HT-29 colon cancer cells, in which the exposure of cells to high concentrations of H_2_O*_2_* produced the activation of AMPK, as suggested by the increased phosphorylation of acetyl-CoA carboxylase (ACC) at Ser79, a well-established target of the protein complex [19]. Even in this case, the effects of ROS were associated with the increase in AMP concentration.

Another possible mechanism for the ROS-mediated activation of AMPK was hypothesized in 2009, when Emerling and colleagues demonstrated that under hypoxic conditions, mitochondrial ROS can activate AMPK in MEF cells, without increasing the relative level of AMP. These data suggest that the association between ROS and AMPK regulation may be AMP-independent, implying that different mechanisms can lead to the activation of AMPK on an increase in ROS levels [20]. Accordingly, it was subsequently demonstrated that in human 143B osteosarcoma cells, the hypoxia-mediated increase in ROS levels causes a rise in the intracellular concentration of calcium. This effect leads to the activation of CaMKKβ, which phosphorylates AMPK, stimulating its activity [21].

Overall, these pieces of information point toward ROS as positive regulators of AMPK activity. It is important to underline that exogenous cell treatment with H_2_O_2_ was shown to cause a rapid enhancement of AMPK activity (within 5 min of H_2_O_2_ treatment), which decreased to the basal level after about one hour of cell exposure to the oxidative insult, suggesting that the effect of ROS on the protein may depend on the duration of the stimulus [17]. This result may reflect the intrinsic ability of AMPK to perceive intracellular environmental cues and modulate its activity according to the type, intensity, and duration of the stimulus. Even though the effects of ROS on AMPK activity are consistent in all the aforementioned studies, it is worth mentioning that different pathways have been proposed through which the ROS-mediated regulation of AMPK may be achieved. The relative importance of one mechanism compared to the other may be cell- or context-dependent.

A recent work made this picture more complicated. In fact, in specific circumstances, the activity of AMPK has been shown to be downregulated upon ROS level enhancement [22]. Indeed, it has been demonstrated both in vivo and in vitro that oxidative stress caused by the increase in glucose levels in skeletal muscle is followed by a reduction in AMPK steady state and activity [22]. More specifically, it was shown that upon glucose treatment, the increase in ROS concentration led to the alteration of two different signaling cascades, which eventually resulted in AMPK downregulation. On the one hand, an oxidative environment promoted the dissociation of AMPK from the upstream kinase LKB1, leading to a decrease in phosphorylation at the level of Thr172 and the consequent inactivation of the protein complex. On the other hand, ROS promoted the activity of protein kinase B (AKT), which is a negative regulator of AMPK. Indeed, AKT phosphorylated both Ser485 and Ser491 of the AMPKα subunit, and these posttranscriptional modifications increased the proteasomal degradation of the protein complex [22]. Overall, this work demonstrates that the effects of ROS on the activity of AMPK may vary in different tissues and models, leading to opposite results. Moreover, it suggests that the AMPK function may be regulated at different levels, not only through the posttranscriptional modulation of its activity but also through the control of its rate of transcription and degradation (Figure 1).

## 4. ROS and AMPK: Direct Interaction

Altogether, the data presented so far demonstrated that ROS may indirectly modulate AMPK activity. However, some recent reports showed that AMPK can undergo oxidative modifications at the level of the α subunit as a consequence of high concentrations of H_2_O_2_. In HEK293T cells, several AMPK cysteine residues, like Cys299, Cys304, and Cys312, have been demonstrated to be rapidly oxidized and to be the target of oxidative-dependent posttranscriptional modifications, such as S-glutathionylation. These alterations were accompanied by the hyperphosphorylation of Thr172. It is likely that cysteine oxidation leads to conformational changes in AMPK that increase the ability of the protein to be phosphorylated by its upstream kinases. Importantly, the effect of cysteine oxidation was shown to anticipate the variation in the AMP:ATP ratio observed upon H_2_O_2_ treatment, suggesting that the direct interaction between ROS and AMPK may be highly relevant for the fast response of the protein complex to oxidative conditions [23].

Another research work revealed that in cardiomyocytes cysteine oxidation leads to the opposite outcomes. More precisely, Cys130, Cys174, and Cys490 were shown to be the most affected residues by oxidation after cell exposure to H_2_O_2_. The modification of these amino acids was suggested to affect AMPK conformation, decreasing the rate of AMPK phosphorylation at the level of Thr172 and the activity of the protein. In fact, in this cell type, AMPK oxidation induced the aggregation of the α subunit, leading to the formation of a complex structure unable to be phosphorylated by the AMPK upstream kinases, resulting in the inhibition of the protein. This mechanism does not seem to affect the basal level of AMPK phosphorylation, but determines the inability of the protein to be further phosphorylated and, therefore, activated [24].

Overall, the data described here confirm the complexity of AMPK regulation. The different results obtained exploiting different models suggest that the modulation of this protein complex may be highly cell and tissue specific. However, it is important to underline that the discrepancies in the results may also be explained by differences in the reactive molecules. Indeed, the characterization of the effect of specific ROS species is still poorly developed: most of the experiments investigating the role of oxygen species rely on exogenous treatment with H_2_O_2._ In contrast, analyses of endogenous ROS are usually performed with reporter molecules that are not able to clearly discriminate among the different oxidative agents and are used to analyze the general level of ROS. In this picture, we cannot exclude the possibility that the relative concentration of different ROS may be specifically affected by different stimuli, leading, eventually, to distinct effects.

It is also crucial to highlight the fact that each AMPK subunit may be found in different isoforms. For instance, the α subunit, which appears to be the most relevant for the ROS-dependent regulation of the protein complex, is transcribed by two genes in two different isoforms, α1 and α2. Although the relative importance and the differences between the two isoforms are not yet understood, it has been shown that while the α1 isoform is ubiquitously expressed, the α2 isoform is enriched in skeletal and cardiac tissues [13]. It is interesting to note that the analyses performed in these latter experimental models detected a negative correlation between ROS levels and AMPK activation. These observations suggest that the presence of a specific isoform in the AMPK complex may influence the ability of the protein to respond to oxidative stress and may determine a differential activation of the protein. It would be highly relevant to characterize the differences among the AMPK subunits and to understand whether and how they promote specific activities and regulations.

We also need to mention that while AMPK is frequently considered a unique entity, it has been recently demonstrated that several distinct subcellular pools of the protein are localized at specific cell compartments, such as mitochondria, lysosomes, or at the level of the nucleus [25]. These AMPK subpopulations might be highly specialized to sense variation in their microenvironment, thus promoting the most appropriate response at the level of their subcellular region. Moreover, the different AMPK subpopulations have been proposed to be independently regulated and to be able to influence different pathways [25,26]. In this picture, we can speculate that, since mitochondria are the major contributors to ROS production, the mitochondrial pool of AMPK may be the most susceptible to variation in ROS concentration and the fastest pool to respond to oxidative stress. For this reason, it would be crucial to focus on the effect of ROS on this specific fraction of the protein and characterize the response of this pool to alterations in ROS levels (Figure 1).

## 5. Reactive Nitrogen Species and AMPK

Similar to ROS, reactive nitrogen species (RNS) are highly reactive molecules originating from the oxidation of nitric oxide that comprise peroxynitrite, nitroxyl, and nitrosonium cation [27]. Like ROS, these molecules are able to react with and influence the activity of other cellular biomolecules; for this reason, RNS may represent important signaling entities, but when their concentration exceeds the physiological level, they can lead to nitrosative stress. With regard to proteins, RNS can promote post-transcriptional modifications, such as the nitrosylation of sulfhydryls (S-nitrosylation) or metals, and the nitration of tyrosine residues [27].

In contrast to ROS, the effects of RNS on AMPK activity have not been completely elucidated yet. Indeed, only the indirect RNS-mediated modulation of AMPK have been proposed, suggesting that AMPK becomes activated upon an increase in RNS levels. In MCF-7 cells exogenously exposed to nitric oxide, the stimulation of AMPK was associated with the activation of ATM, which is a stress response protein whose function is promoted upon DNA damage as well as oxidative and nitrosative stress [28]. ATM promotes LKB1 function that phosphorylates AMPK. Importantly, the activation of the AMPK pathway in this context results in an increase in autophagy, corroborating the idea that ROS/RNS can strongly influence autophagic clearance activity through the regulation of AMPK [28].

More recently, the homolog of AMPKα in the yeast Schizosaccharomyces pombe was shown to be phosphorylated upon an increase in RNS levels due to the activity of the upstream kinase CaMKKβ [29]. The mechanism for the RNS-dependent activation of CaMKKβ was not characterized, but it may involve the increase in Ca^2+^ levels, one of the most important modulators of the kinase activity.

It is important to underline that the direct modulation of AMPK determined by RNS has not been addressed yet; however, these molecules are able to promote post-transcriptional modification of proteins that can affect their function [30]. Therefore, we can hypothesize that RNS may also promote a direct effect on AMPK activity. This issue should be investigated in the future to fully understand the link between RNS and AMPK.

Nitric oxide can be considered the main RNS molecule, as it represents the starting point for the formation of other nitrogen species. It is produced by the catalytic activity of nitric oxide synthases (NOS), from L-arginine and oxygen. Three isoforms of NOS have been described, which have different mechanisms of activation and individual patterns of expression [31]. At the subcellular level, NOS can be found in different compartments, suggesting that RNS generation may take place at different cellular levels [5]. The production of nitric oxide is essential since it serves as a signaling molecule, and mitochondria have been demonstrated to represent targets of nitric oxide, where the S-nitrosation of mitochondrial proteins can occur, resulting in the RNS-dependent regulation of their functions [32]. Accordingly, there is evidence suggesting the presence of a mitochondrial NOS, which promotes the generation of nitric oxide at the mitochondrial level, although a general agreement on the identity of this mitochondrial NOS is still under debate [32]. For these reasons, mitochondria may represent an important source of RNS that can affect AMPK and, therefore, its downstream targets.

Overall, these data indicate that RNS can lead to the activation of AMPK. However, the pieces of information on this topic are still poorly developed. We can hypothesize that, like ROS, the RNS-mediated regulation of AMPK could also depend on the tissue or the cell type, or even vary at the level of different subcellular compartments. More research efforts would be needed to clarify these questions and characterize in detail how AMPK is regulated by RNS.

## 6. AMPK and Autophagy

As reported in the previous paragraphs, a strong link between oxidative stress and AMPK activity exists. By influencing the function of this protein complex, ROS can participate in the regulation of several important cellular processes. Moreover, with ROS production being highly subjected to variations in mitochondrial function, through the involvement of AMPK, ROS can participate in the transduction of signals relative to the mitochondrial homeostatic state. In turn, AMPK can modulate crucial molecular pathways which ensure the maintenance and regulation of the overall cellular energetic balance, in response to mitochondrial-derived signals [6]. Indeed, AMPK represents a key hub involved in crucial signaling pathways and its kinase activity influences cell growth, cell differentiation, mitochondrial dynamics, and autophagy [14].

Autophagy, in particular, is one of the major cellular processes affected by AMPK [13], which regulates this catabolic machinery through different molecular mechanisms.

Autophagy represents the main intracellular degradative process and is performed through the recruitment of cell debris, such as damaged organelles and dysfunctional misfolded proteins, within double-membrane vesicles called autophagosomes, which deliver their content to the lysosomes. The fusion between autophagosomes and lysosomes leads to the generation of autolysosomes, characterized by an acidic lumen and by the presence of a broad variety of hydrolytic enzymes. These vesicles represent the sites where autophagic cargoes are degraded. Then, the breakdown products are recycled to sustain metabolic functions [33].

One of the best-characterized signaling pathways that allows the AMPK-mediated modulation of autophagy relies on the activity of the mechanistic target of rapamycin complex 1 (mTORC1). mTORC1 is a protein complex whose kinase activity strongly inhibits autophagy in response to environmental cues [34]. AMPK has been demonstrated to act as a negative regulator of mTORC1, resulting in autophagic induction. The interaction between these two proteins is mediated by the tuberous sclerosis complex (TSC), which is phosphorylated and activated by AMPK and, in turn, represses mTORC1 [35]. In addition, the AMPK-dependent inhibition of mTORC1 can be the result of a direct interaction. Actually, AMPK has been observed to phosphorylate the mTORC1 subunit RAPTOR, causing the inactivation of the protein complex and the consequent increase in autophagic activity (Figure 2) [36].

It is worth mentioning that, since mTORC1 affects autophagy through different pathways, via mTORC1 inhibition AMPK can also modulate autophagy by acting at different levels. Indeed, two of the best-characterized targets of mTORC1 are the transcription factor EB (TFEB) [37], which tunes the expression of proteins involved in the autophagic pathway [38], and the unc-51-like autophagy activating kinase 1 (ULK1) [39], which is essential for the initiation of the first steps of the autophagic process [40]. As a consequence, the AMPK-mediated inhibition of mTORC1 results in the activation of these two protein-related pathways, leading to the induction of autophagy. Interestingly, AMPK has also been demonstrated to directly phosphorylate both TFEB and ULK1, promoting their activation (Figure 2) [41,42].

Altogether, these data confirm the strong involvement of AMPK in the autophagic pathway. Moreover, they imply that the mechanisms of autophagic regulation mediated by AMPK are multiple and, sometimes, redundant, likely to ensure a more precise and tuned effect of the protein on this process.

In this frame, it appears clear that ROS, through the involvement of AMPK, can participate in the regulation of the autophagic pathway (Figure 2). Indeed, oxidative stress has been associated with autophagy in different research works. For example, in rat primary cardiac myocytes it has been observed that exposure to H_2_O_2_ leads to the induction of autophagic flux [43]. Similar results were obtained in HEK293T and Hela cells, in which an increase in autophagic activity was linked to enhanced levels of superoxide radicals [44,45]. More recently, the ROS-mediated alteration of the autophagic process has been demonstrated in vivo using *Drosophila melanogaster.* In this model, the increase in ROS levels caused by the absence of the Parkinson’s Disease-related protein DJ-1 led to the accumulation of lysosomes with the concomitant reduction in the number of autolysosomes. Importantly, ROS scavenging was described to be sufficient to rescue the altered phenotypes, clearly demonstrating the link between oxidative stress and autophagic regulation [46].

Whether ROS induce or repress the autophagic flux is still not clear and the results in different models are not always consistent, likely due to the fact that ROS may trigger cell-specific pathways and that, as mentioned before, the analysis of the effects mediated by different ROS is still lacking [47]. These discrepancies may also be determined by ROS concentrations. Accordingly, in SH-SY5Y cells, low concentrations of H_2_O_2_ were shown to promote TFEB activation and autophagy induction, while higher amounts of H_2_O_2_ caused TFEB repression and the inhibition of the autophagic flux [48]. These results might reflect the differences in the effects promoted by ROS on AMPK, which, as already emphasized, differ among experimental models and upon different conditions. These data highlight the need to better investigate how ROS impact on AMPK activity and, in turn, autophagy.

It is worth mentioning that the intersection between ROS and autophagy has been thoroughly analyzed in the context of neurodegeneration. In this frame, the manipulation of autophagic activity has been proposed as an effective strategy to alleviate oxidative stress in patients suffering from neurodegenerative diseases, confirming the presence of two-way communication that links ROS to the autophagic pathway [49].

## 7. The Link between ROS and DNA Damage and Repair: The Role of AMPK

As thoroughly reported in this paper, ROS are reactive molecules which have a broad spectrum of targets. Among them, DNA has been widely investigated as one of the biomolecules that may be affected by oxidative damage [50]. The effects of ROS on DNA comprise the loss of nucleobases, base damage, and single and double DNA strand breaks [51]. Importantly, mitochondria being the major intracellular source of ROS, mitochondrial DNA appears to be one of the primary targets of oxidative mutations [51]. In this scenario, it is clear that high levels of ROS may critically impact both mitochondrial and nuclear DNA stability, potentially leading to deleterious mutations that can undermine cell homeostasis and viability.

To avoid the detrimental effects of DNA damage cells can implement key mechanisms to repair DNA, and the induction of the AMPK pathway has been proven to play a key role in the process of DNA damage repair. AMPK activation upon DNA damage has been associated with the induction of the Ataxia-telangiectasia mutated (ATM) protein [52]. ATM is a stress response protein kinase whose function is promoted upon the detection of DNA double-strand breaks. ATM promotes AMPK activation through at least two mechanisms; on the one hand, it phosphorylates and activates the AMPK upstream kinase LKB1, and on the other hand this protein has been observed to directly increase AMPK phosphorylation at Thr172 [52,53]. Importantly, ATM can also be activated by oxidation, suggesting that the increase in the levels of ROS may induce the function of this protein [54]. The upregulation of the AMPK pathway may promote DNA damage repair by stimulating different signaling cascades. First, by inhibiting mTORC1, AMPK induces the expression of the nuclear factor erythroid 2-related factor 2 (NRF-2). NRF-2 is a transcription factor involved in the antioxidant response. Among the proteins upregulated by NRF2 activity, 8-oxoguanine glycosylase (OGG1) is essential in the process of base excision repair [52]. In addition, it has been demonstrated that the activation of AMPK promotes the repair of tumorigenic DNA lesions in a skin tumor mouse model of UVB-induced DNA damage [55]. In this frame, AMPK has been shown to enhance the activity of the DNA-repairing protein xeroderma pigmentosum C (XPC), which participates in global genome nucleotide excision repair [55]. Although the precise molecular mechanisms that mediate the AMPK-dependent activation of XPC have not been fully characterized, it seems that AMPK is not involved in the transcriptional regulation of XPC, but rather in the post-transcriptional activation of the protein [55].

Noteworthily, AMPK can serve as a pro-DNA repair factor thanks to its function as a regulator of the cell cycle. AMPK participates in the regulation of cell growth, not only through the inhibition of mTORC1, which promotes cell cycle arrest [36], but also via the phosphorylation of protein involved in mitotic progression, such as protein phosphatase 1 regulatory subunit 12C (PPP1R12C) and p21-activated protein kinase (PAK2) [53]. Importantly, the eukaryotic cell cycle is regulated by checkpoint mechanisms that allow the progression of this process only when cells are in suitable condition and if DNA is not damaged. In this context, AMPK has been shown to inhibit cell cycle progression, inducing checkpoint mechanisms and facilitating cell survival upon DNA damage [56].

Importantly, the induction of autophagy has also been observed to promote DNA damage repair through several mechanisms that are exhaustively reviewed elsewhere [57] and comprise the enhancement of base excision repair, mismatch repair, and nucleotide excision repair. Therefore, by stimulating the autophagic pathway, AMPK can indirectly promote the repair of damaged DNA.

It is important to underline that the positive effect on AMPK exerted by ROS may serve as important feedback to induce DNA repair mechanisms to prevent and counteract the possible detrimental consequences of oxidative stress on DNA, but also on proteins and other biomolecules. Moreover, as aforementioned, AMPK activity can increase the expression and the function of antioxidant genes, such as NRF-2, promoting the maintenance of balanced and stable ROS levels [58].

Altogether, these data demonstrate that the functional link between ROS and AMPK does not only influence autophagy, but may also participate in the regulation of other important processes, ensuring the control of overall cell homeostasis.

## 8. Future Challenges in the Study of AMPK Activation by ROS

While AMPK appears to play a central role in autophagy by controlling the process at different levels, it is important to underline the fact that AMPK is not the only modulator of autophagy, and that ROS may have different targets that potentially participate in the regulation of this pathway. Indeed, autophagy is crucially involved in maintaining cell homeostasis, and its functions are finely tuned by a complex network of proteins that allow this process to promptly respond to a plethora of intracellular and extracellular stimuli [59]. For this reason, the assumption that there is always a direct correlation between AMPK activity and autophagy regulation appears to be a simplified vision. Accordingly, it has been recently suggested that the molecular pathways triggered by AMPK may be highly cell-specific. For example, it has been observed that the activation of AMPK in neuronal cells is not associated with the increase in autophagic degradation, suggesting that in this cell type the protein complex may have a limited role in the induction of autophagy [60]. Therefore, it will be crucial to precisely identify the contribution of oxidative stress to the AMPK function and of the pathways affected by its activity, with particular attention paid to the effect of specific ROS molecules, and in different cells.

Another essential aspect that needs to be taken into account when investigating the role of AMPK in autophagy is that the protein may also have different properties in the same cell, in accordance with its subcellular localization [25,26]. For instance, a recent work demonstrated that a subpopulation of AMPK localized at the mitochondria was highly specialized to sense variation in mitochondrial dynamics, inducing responses that tended to restore mitochondrial homeostasis with high spatial specificity [61]. As previously mentioned, it is likely that the mitochondrial AMPK pool may be the first to be activated and the most important fraction of the protein affected by altered ROS levels. By analogy, we could expect that autophagy and lysosomal functions may be mainly subjected to the activity of the lysosomal AMPK fraction. Indeed, at the lysosomal level, AMPK may easily target proteins that are involved in the modulation of the autophagic pathway, such as mTORC1 and TFEB [25].

Altogether these considerations suggest that targeting AMPK to impact autophagy using generic drugs, such as metformin [62], may be beneficial on one side, but could also cause unpredictable alterations, ultimately affecting other cellular homeostatic processes. If feasible, in the future, specific drugs targeting specific AMPK pools and directed toward specific cell types and tissues should be developed.

## 9. Conclusions

Overall, the data presented in this review emphasize that, through the modulation of AMPK, mROS may be involved in the transduction of information from mitochondria to other cell compartments, influencing both lysosomal activity and the autophagic flux.

Nevertheless, as pointed out in the previous paragraph, the picture is quite intricate and presents some major challenges. Thus, the investigation of the functions of AMPK, not as a unique molecule, but as a combination of different proteins with subcellular-, cellular- and tissue-specific activities, would be of great help. Therefore, to deeply characterize the effect of oxidative stress on AMPK and, consequently, on the autophagic pathway, it would be advisable to evaluate the specific response to altered ROS levels, not only of the mitochondria-localized AMPK pool, but also of the lysosomal-localized AMPK fraction, given the role of lysosomes in the autophagic process. This will allow us to characterize the main pathways regulated by the different fractions of the protein and to understand how they can be involved in the overall lysosomal and autophagic activity. Moreover, this will allow us to better understand how AMPK pools impact on autophagy, and possibly also in different cell types and tissues.

## Figures and Tables

**Figure 1 antioxidants-12-01406-f001:**
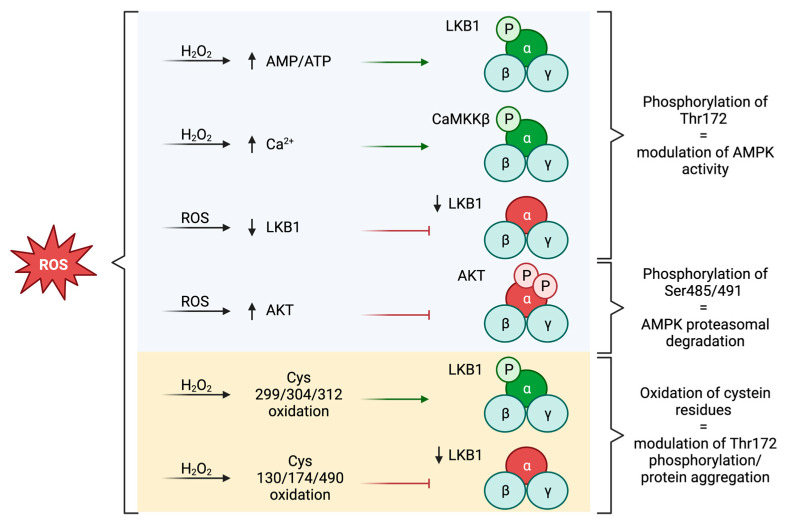
ROS-mediated regulation of AMPK. The indirect mechanisms of regulation of AMPK promoted by ROS are reported in the blue box. The yellow box represents the mechanisms by which ROS can directly affect AMPK activity. ROS affect AMPK activity mainly through the modulation of the α subunit, which is represented in green when the activity of the protein is promoted and in red when AMPK is inhibited by ROS. (Created with BioRender.com, accessed on 7 June 2023).

**Figure 2 antioxidants-12-01406-f002:**
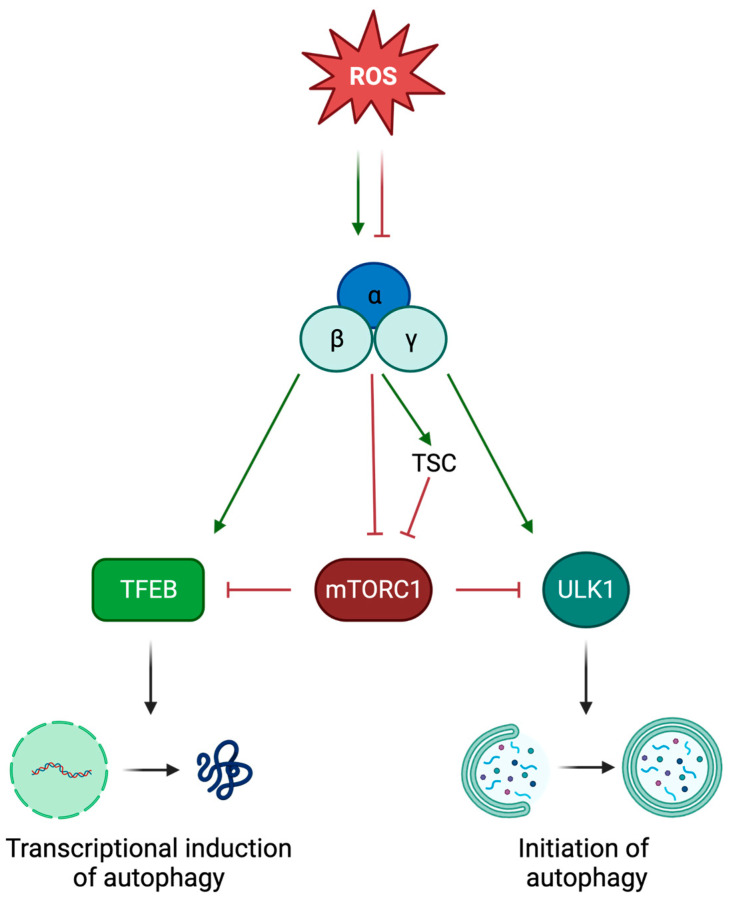
ROS can affect autophagic activity. The diagram represents the molecular mechanisms that allow ROS to influence the autophagic pathway, through the regulation of AMPK (created with BioRender.com, accessed on 7 June 2023).

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
