# Peer review of "Linking ROS Levels to Autophagy: The Key Role of AMPK"

_antioxidants, 2023, doi:10.3390/antiox12071406_

Round 1

Reviewer 1 Report

In the review article entitled: Linking ROS Levels to Autophagy: the Key Role of AMPK authors have taken into consideration the crosstalk between mitochondria and lysosomes. The above has been discussed in light of autophagy. Also, the role of reactive oxygen species in the AMPK activation process was discussed. Even though the article is interesting for audiences from different fields, readable, and well-written I have some critical remarks which should be answered before publication. 

1) The authors did not discuss other reactive species like nitrogen

2) The role of ROS and therefore AMPK in the DNA damage formation have been completely omitted

3) The meaning of AMPK in nucleic acids repair processes must be focused

4) From the nutritional point, autophagy should be discussed in the different types of feeding.

In conclusion, the major correction should be made before publication. It would be a wonder if the medicinal point of AMPK and autophagy will be shown

Reviewer 2 Report

The manuscript entitled “Linking ROS Levels to Autophagy: the Key Role of AMPK” by Francesco Agostini et al. carefully reviewed the molecular mechanisms through which reactive oxygen species associated with AMP-activated protein kinase, the interconnection role of the protein and the signaling pathways that allow AMPK to participate in the autophagic process. The premise of the work is very interesting, however in its present version, the manuscript requires several significant areas of improvement before consideration for publication. In my opinion major improvements are required

It is not clear if mitochondria-derived ROS activate AMPK and this point needs to be shown. Also does the protein maintain cellular metabolic homeostasis through regulation of mitochondrial ROS? The authors should stress more as it is difficult for readers who don't have a strong basic sciences foundation to understand this section.

I wonder if authors would provide some bullet points or a distinct paragraph addressing some major points and challenges of some demanding questions of the discussed area.

Perhaps the authors could present some specific targets for future studies in conclusions section as briefly mentioned in the abstract.

 Minor editing of English language required

Round 2

Reviewer 1 Report

The authors have provided the required correction adequate to my question. Therefore, I can recommend this article for publication in the Antioxidants journal.

Reviewer 2 Report

The revised manuscript addresses my comments and suggestions.

English language is fine